# NButGT Reinforces the Beneficial Effects of Epinephrine on Cardiac Mitochondrial Respiration, Lactatemia and Cardiac Output in Experimental Anaphylactic Shock

**DOI:** 10.3390/ijms25063316

**Published:** 2024-03-14

**Authors:** Walid Oulehri, Antoine Persello, Angelique Blangy-Letheule, Charles Tacquard, Bertrand Rozec, Anne-Laure Charles, Bernard Geny, Benjamin Lauzier, Paul Michel Mertes, Olivier Collange

**Affiliations:** 1Pôle Anesthésie, Réanimation Chirurgicale, Hôpitaux Universitaires de Strasbourg, 67091 Strasbourg, France; charlesambroise.tacquard@chru-strasbourg.fr (C.T.); paul-michel.mertes@chru-strasbourg.fr (P.M.M.); olivier.collange@chru-strasbourg.fr (O.C.); 2Faculté de Médecine de Strasbourg, UR 3072 Institut de Physiologie, FMTS (Fédération de Médecine Translationnelle de Strasbourg), Université de Strasbourg, 67091 Strasbourg, France; anne.laure.charles@unistra.fr (A.-L.C.); bernard.geny@chru-strasbourg.fr (B.G.); 3Institut du Thorax, Université de Nantes, CNRS, INSERM, 44000 Nantes, France; antoine.persello@gmail.com (A.P.); angelique.blangy-letheule@univ-nantes.fr (A.B.-L.); bertrand.rozec@chu-nantes.fr (B.R.); benjamin.lauzier@univ-nantes.fr (B.L.); 4Service de Physiologie et d’Explorations Fonctionnelles, Pôle de Pathologie Thoracique, Hôpitaux Universitaires de Strasbourg, 67091 Strasbourg, France

**Keywords:** NButGT, O-GlcNAcylation, cardiac protection, mitochondrial respiration, anaphylactic shock

## Abstract

Anaphylactic shock (AS) is the most severe form of acute systemic hypersensitivity reaction. Although epinephrine can restore patients’ hemodynamics, it might also be harmful, supporting the need for adjuvant treatment. We therefore investigated whether NButGT, enhancing O-GlcNAcylation and showing beneficial effects in acute heart failure might improve AS therapy. Ovalbumin-sensitized rats were randomly allocated to six groups: control (CON), shock (AS), shock treated with NButGT alone before (AS+pre-Nbut) or after (AS+post-Nbut) AS onset, shock treated with epinephrine alone (AS+EPI) and shock group treated with combination of epinephrine and NButGT (AS+EPI+preNBut). Induction of shock was performed with an intravenous (IV) ovalbumin. Cardiac protein and cycling enzymes O-GlcNAcylation levels, mean arterial pressure (MAP), heart rate, cardiac output (CO), left ventricle shortening fraction (LVSF), mitochondrial respiration, and lactatemia were evaluated using Western blotting experiments, invasive arterial monitoring, echocardiography, mitochondrial oximetry and arterial blood samples. AS decreased MAP (−77%, *p* < 0.001), CO (−90%, *p* < 0.001) and LVSF (−30%, *p* < 0.05). Epinephrine improved these parameters and, in particular, rats did not die in 15 min. But, cardiac mitochondrial respiration remained impaired (complexes I + II −29%, *p* < 0.05 and II −40%, *p* < 0.001) with hyperlactatemia. NButGT pretreatment (AS+pre-Nbut) efficiently increased cardiac O-GlcNAcylation level as compared to the AS+post-Nbut group. Compared to epinephrine alone, the adjunction of NButGT significantly improved CO, LVSF and mitochondrial respiration. MAP was not significantly increased but lactatemia decreased more markedly. Pretreatment with NButGT increases O-GlcNAcylation of cardiac proteins and has an additive effect on epinephrine, improving cardiac output and mitochondrial respiration and decreasing blood lactate levels. This new therapy might be useful when the risk of AS cannot be avoided.

## 1. Introduction

Anaphylactic shock (AS) is the most severe, life-threatening form of acute systemic hypersensitivity reaction [1]. The pathophysiology of AS classically includes a complex mixed shock combining a vasoplegic and a hypovolemic component [2] associated with a genuine cardiac dysfunction [3,4]. Epinephrine is the cornerstone of AS treatment but anaphylactic reactions can lead to refractory shock unresponsive to epinephrine [5] and death [6]. On the other hand, epinephrine can also drive deleterious effects by promoting the onset of stress cardiomyopathy [7,8,9,10]. Using an experimental model of AS, our team has already shown that epinephrine can stimulate cardiac function by increasing inotropism and shortening the fraction of an empty heart [4]. In this case, epinephrine might likely worsen the initial clinical condition of AS. Thus, epinephrine treatment for AS would benefit from being combined with adjuvant treatment, either to improve its performance or to decrease the total administrated dose, aiming to limit the deleterious effects while preserving its beneficial effects.

We previously described in an animal model of AS that cardiac dysfunction during AS was associated with mitochondrial respiration dysfunction [11]. O-linked β-N-acetylglucosamine (O-GlcNAcylation) is a reversible post-translational modification of the glycosylation family [12,13]. Increased O-GlcNAcylation is protective against acute cellular stress probably by enhancement of mitochondrial respiration and through a decrease in oxidative stress production [14]. Protein O-GlcNAcylation requires uridine disphosphate N-acétylglucosamine (UDP-GlcNAc) as a substrate, which is the final product of the hexoamine biosynthesis pathway [15]. The addition of the UDP-GlcNAc moiety to a serine or threonine of a target protein is allowed by an enzyme, O-GlcNAc transferase (OGT), and the moiety is removed by O-GlcNAcase (OGA). Several mitochondrial targets related to the respiratory chain were highlighted to explain the relation between O-GlcNAcylation and mitochondrial function: NADH dehydrogenase and ubiquinone reductase modules of complex I, hydrophilic head of complex II, a matrix-facing subunit of complex III and F1 subunit of complex V [15]. Increasing O-GlcNAcylation levels seems to be a potent target therapy in situations of intense cellular stress, such as during ischemia–reperfusion, septic [16], hypovolemic shock [17] and/or myocardial infarction [18]. The NButGT, a highly selective competitive inhibitor binds the active site of OGA to block the removal of UDP-GlcNAc moieties [19] and significantly elevates O-GlcNAcylated protein level [19]. NButGT and stimulation of O-GlcNAcylation have already been studied in experimental models of septic and hypovolemic shock, but never in a model of anaphylactic shock.

The aim of this study was to determine (1) whether NButGT treatment increases O-GlcNAcylation in cardiac tissue and (2) whether NButGT as an adjuvant treatment of epinephrine can improve hemodynamic, cardiac function and cell metabolism in an experimental model of lethal anaphylactic shock.

## 2. Results

### 2.1. First-Step Study: NButGT Efficiently Increases Total Cardiac O-GlcNAcylation Protein Only When Administered before AS

The global O-GlcNAcylation level significantly increased at T15 min in AS+pre-Nbut group as compared with CON-T15 group (2.50 ± 0.15 vs. 1.00 ± 0.06, *p* < 0.0001, respectively), AS group (2.50 ± 0.15 vs. 1.25 ± 0.17, *p* < 0.0001, respectively) and AS+post-Nbut group (2.50 ± 0.15 vs. 1.16 ± 0.07, *p* < 0.0001, respectively) (Figure 1A). OGA and OGT expression in cardiomyocytes did not differ between groups (Figure 1B,C). These results were obtained without epinephrine administration. Based on the O-GlcNAcylation increase in the AS+pre-Nbut group, this protocol of NButGT administration was selected to evaluate the interaction between NButGT and epinephrine to treat AS.

#### Cardiac Mitochondrial Respiration

We observed a decrease in oxidative phosphorylation (OXPHOS) through complex II (CII OXPHOS state) in the AS group as compared to the CON-T15 group (Figure 2): −36%, 133 ± 15 vs. 209 ± 15 pmol/s/mg, *p* < 0.05, respectively. Moreover, pretreatment with NButGT (AS+pre-Nbut group) improved significantly mitochondrial respiration in the CII OXPHOS state as compared to the AS group (216 ± 30 vs. 133 ± 15 pmol/s/mg, *p* < 0.05, respectively), to reach value of the CON-T15 group (216 ± 30 vs. 209 ± 15 pmol/s/mg, *p* = 0.97; in the AS+pre-Nbut and CON-T15 groups, respectively).

### 2.2. Second-Step Study: Effects of NButGT Pretreatment on Hemodynamics, Cardiac Function and Mitochondrial Respiration

#### 2.2.1. Hemodynamic Parameters

MAP remained constant (128 ± 1 mmHg) during the whole experimentation period in the CON group (Figure 3A).

In the AS, AS+EPI, and AS+EPI+preNBut groups, ovalbumin (OVA) injection resulted in a similar severe and early decrease in MAP (Figure 3A). In the AS group, MAP continued to decrease until 15 min, the end of the experiment for this group, to reach 31 ± 1 mmHg. In the AS+EPI and AS+EPI+preNBut groups, MAP increased progressively in response to the epinephrine infusion to reach a maximum value at T20 and T25, respectively: 119 ± 5 mmHg and 120 ± 5 mmHg (*p* = NS).

Total epinephrine infusion dose was not different between the AS+EPI and AS+EPI+preNBut groups: 889 ± 67 mcg/kg vs. 879 ± 45 mcg/kg, *p* = 0.79, respectively (Figure 3B).

HR remained stable during the whole experiment in the CON group (421 ± 2/min) (Figure 3C). Within the first minutes of shock, HR decreased significantly in the AS, AS+EPI and AS+EPI +preNBut groups (compared to the CON group, *p* < 0.0001). From T0 to T15 min, HR decreased and remained lower in the AS group as compared to the CON group (*p* < 0.0001). HR increased at T2.5 in the two groups treated with epinephrine (AS+EPI and AS+EPI+preNBut) to reach the values of the CON group. After T5, no significant difference was observed in HR values between the CON, AS+EPI and AS+EPI+preNBut groups.

#### 2.2.2. Cardiac Function

We calculated CO values using the following formula, CO = SV × HR, where SV was stroke volume. SV was estimated from LVEDD (left ventricle end-diastolic diameter) and LVESD (left ventricle end-systolic diameter) measurements by the simplified Teichholz formula [20]. This formula is currently used in clinical echocardiography to estimate SV.

LVEDD remained constant during the whole experiment in the CON group (5.8 ± 0.1 mm) (Figure 4A). LVEDD immediately decreased in the other groups to reach minimal values between T5 and T15 (Figure 4A). In both epinephrine-treated groups (AS+EPI and AS+EPI+preNBut), LVEDD values increased progressively from T10, without ever reaching the value of the control group. From T32.5 to T60 min, LVEDD increased more markedly in the AS+EPI+preNBut group compared to the AS+EPI group (*p* < 0.05). During this period, mean LVEDD was 3.4 ± 0.1 mm in the AS+EPI group and 4.3 ± 0.1 mm in the AS+EPI+preNBut group.

LVESD remained constant during the whole experiment in the CON group (3.1 ± 0.1 mm) (Figure 4B). LVESD immediately decreased in the other groups to reach minimal values between T5 and T15 (Figure 4B). In both epinephrine-treated groups (AS+EPI and AS+EPI+preNBut), LVESD values increased slightly from T30 to the end of the experiment, without ever reaching the value of the control group. LVESD increase was not statistically different between AS+EPI and AS+EPI+preNBut groups during the whole experiment.

LVSF (left ventriclular shortening fraction) remained constant during the whole experiment in the CON group (mean LVSF 43 ± 1%) (Figure 5A). In the AS group, the LVSF decreased significantly from T0 to T15; at T15, the LVSF value was significantly lower than in the CON group: 31 ± 3% vs. 47 ± 6%, *p* < 0.05, respectively. Compared to the CON group, we observed a significant increase in LVSF in the AS+EPI group (*p* < 0.05) from T5 to T22.5 min and in the AS+API+preNBut group (*p* < 0.0001) from T5 to T60 min. Indeed, from T22.5 to T60 min in the AS+EPI group, LVSF gradually decreased until it reached the value of 59 ± 14% at T60. In the AS+EPI+preNBut group, LVSF did not decrease and remained at a high level throughout the experiment. Moreover, from T35 min until the end of the experiment, LVSF was significantly higher in the AS+EPI+preNBut group as compared to the AS+EPI group (*p* < 0.05) (Figure 5A).

Cardiac output (CO) remained relatively constant during the whole experiment in the CON group (62 ± 3 mL/min) (Figure 5B). OVA injection resulted in a profound and rapid decrease in CO in the AS group, as compared to the CON group (*p* < 0.001), to reach the minimal value of 5 ± 3 mL/min at T15 min. In the AS+EPI and AS+EPI+preNBut groups, CO decreased to reach minimal values of 8 ± 1 mL/min at T10 min and 7 ± 2 mL/min at T7.5 min, respectively. From T10 min, epinephrine alone or in association with NButGT enhanced poorly CO. However, from T40 min to the end of the experiment CO was significantly increased in the AS+EPI+preNBut group, as compared with the AS+EPI group (*p* < 0.05) (Figure 5B). During this period, CO was 35 ± 1 mL/min and 15 ± 1 mL/min in the AS+EPI+preNBut and AS+EPI groups, respectively.

#### 2.2.3. Cardiac Mitochondrial Respiration

Concerning mitochondrial respiration, OXPHOS was significantly impaired in the AS+EPI groups as compared to the control group through complexes I+II (−29%, *p* < 0.05) and II (−40%, *p* < 0.001) at T60 min (Figure 6 and Table 1). Pretreatment with NButGT partially restored the activity of the respiratory chain complexes. As compared with the AS+EPI group: (1) the activity of complex I alone (CI OXPHOS state) was enhanced in the AS+EPI+preNBut group (+38%, *p* < 0.05); (2) the combined activity of complexes I and II (CI+II OXPHOS state) was completely restored in the AS+EPI+preNBut group; and (3) the activity of complex II alone (CII OXPHOS state) was restored to +35%, *p* < 0.01 in the AS+EPI+preNBut group. However, in the AS+EPI+preNBut group, the activity of complex II alone remained lower than the CON group (−20%, *p* < 0.05).

At the end of the experiment (T60 min), arterial lactatemia (Figure 7) was significantly higher in the AS+EPI and AS+EPI+preNBut groups as compared to the CON group (*p* < 0.0001). But, in the AS+EPI+preNBut group, lactatemia was lower as compared to the AS+EPI group: 3.6 ± 0.2 vs. 4.8 ± 0.4 mmol/L, *p* < 0.05, respectively.

## 3. Discussion

Besides confirming the gravity of AS, the main results of this study are that pre-administration of NButGT led to left ventricular proteins O-GlcNAcylation, demonstrating additive beneficial effects to epinephrine characterized by a further increase in cardiac shortening fraction and cardiac output, together with restoration of cardiac mitochondrial respiration and reduced systemic lactatemia.

(1)Characterization of AS and effects of epinephrine alone.

As previously described, in this study, we observed similar alterations after AS onset with major impairments in hemodynamic, cardiac function and cardiac mitochondrial respiration [4,11,21,22]. Thus, without treatment, rats died within 20 min. Indeed, AS involved a dramatic decrease in MAP and HR within the first minutes as compared to the CON group (Figure 3A,C). Concerning cardiac function, we observed that AS impaired LVSF and CO. These results confirm our previous work where a profound decrease in contractility (−25% LVSF) and inotropism (−76% max d*P*/d*t*) after AS were observed [4]. Kuda et al. reported a profound decrease in LV pressure (d*P*/d*t*) caused mainly by reduced coronary blood flow due to coronary spasm in the same model of AS [23]. Triggiani et al. reported that the platelet-activating factor (PAF), a lipid mediator released during anaphylaxis, caused a profound decrease in cardiac contractility [24]. CO impairment was likely related to a decrease in HR and SV. Finally, we observed at T15 min that AS involved a decrease in mitochondrial respiration through complex II (−36%, *p* < 0.05) as compared to the CON-T15 group (Figure 2). Interestingly, these results reinforce the AS-related mitochondrial complex II impairment already observed at T15 min in our previous work [11].

Currently, epinephrine administration associated with IV vascular fillings is the gold standard for the treatment of AS in patients. As expected, in our study, epinephrine associated with vascular filling improved MAP and HR (Figure 3A,C), and treated rats survived until the end of the experiment. Concerning cardiac function, epinephrine improved contractility at the first stage of the experiment. However, from T20 min, we observed a slight decrease in LVSF (Figure 5A). The improvement in CO was not major throughout the whole experiment in the AS+EPI group (Figure 5B), as we observed a stable time course profile from T40 min to the end of the experiment (mean value 15 ± 1 mL/min). Accordingly, Tajima et al. also reported a slight increase in CO when epinephrine was used alone, as compared to the shocked group [22].

Confirming previous data, the mitochondrial respiratory complex I and II activities remained impaired in AS+EPI as compared to the CON group (Figure 6). Hence, epinephrine did not restore mitochondrial respiration, whereas it improved hemodynamics and poor cardiac function.

These results suggest that epinephrine alone is not fully efficient for treating all cell dysfunctions related to AS. Epinephrine might even provoke cardiomyocyte dysfunction, including impaired mitochondrial respiration, which might participate in the epinephrine refractory cases of AS. Li et al. observed in an experimental model of bupivacaine intoxication that beta-adrenergic activation aggravated cardiomyocyte metabolism disorder and contractile dysfunction, mainly through disruption of mitochondrial energy metabolism [25]. Further, epinephrine-induced takotsubo cardiomyopathy was also previously described when epinephrine was used to treat anaphylaxis [8,9,10].

(2)Pretreatment with NbutGT demonstrated additive beneficial effects on top of epinephrine.

We tested whether NbutGT pre and post-treatment, adjuvant to epinephrine, would either improve hemodynamic, cardiac and mitochondrial parameters during AS, or achieve the same beneficial effects with a reduction in the doses of epinephrine required. This might be obtained only if NbutGT is pharmacologically active. In fact, unlike post-shock administration, AS preconditioning with NbutGT enhanced the cardiac protein O-GlcNAcylation level. The increase in O-GlcNAcylation without change in the levels of OGA and OGT expression suggests that the beneficial effects associated with stimulation of O-GlcNAcylation by NbutGT pretreatment are independent of transcriptional regulation. Due to its good efficiency and specificity, NbutGT blocks directly the active site of OGA to maintain a high level of protein O-GlcNAcylation [18]. NbutGT is shown to act as a direct competitive inhibitor of OGA without impacting transcriptome [26]. Dupas et al. [27] did not report any impact of NbutGT on the gene expression profile in an experimental model of septic shock, whereas the O-GlcNAcylome was modified by NbutGT treatment. Thus, the functions of O-GlcNAcylated proteins enable the cell to improve its response to stress. Several other mechanisms to increase O-GlcNAcylation levels have been established:-A different approach to increase O-GlcNAcylation levels is to overexpress OGT by transfection of cultured cells, generating transgenic animals, or infection of cells or tissues with replication-deficient viruses. But, only a modest increase in O-GlcNAcylation was observed with such an approach [28].-Increase in UDP-GlcNAc concentration by glucosamine administration. But, glucosamine has side effects such as a decrease in ATP production [18]-Other specific OGA inhibitors interact directly with the active site of OGA (PUGNAc, Thiamet G). But, they have weaknesses: OGA interaction with poor specificity or too expensive to produce [18].

Several studies have shown that O-GlcNAcylation may have a protective effect in models of hemorrhagic and septic shock. Nöt et al. showed that stimulation of O-GlcNAcylation was associated with a reduction in mortality, inflammation and apoptosis 24 hours after hemorrhagic shock in rats. However, in this study, hemodynamic parameters were not assessed [17]. In a model of septic shock, Ferron et al. [29] observed that the stimulation action of NbutGT is probably at a post-transcriptional level. In this study, the authors administrated NbutGT one hour after septic shock onset. They observed the O-GlcNAcylation level increase without modification of OGA and OGT expression. They suggested that modulation of these enzymes requires a longer period of time. It was precisely because NbutGT pretreatment does not need transcriptional regulation that it might be pertinent in a setting of acute shock occurrence characterizing AS. Indeed, despite observing similar NbutGT-induced improvements in hemodynamic and cardiac function in both septic and anaphylactic shocks, it is crucial to consider the distinct time constants associated with these two types of shocks. In septic shock, cardiovascular and organ dysfunctions may manifest a few hours after the onset of shock, whereas dysfunctions may occur within the first minutes in the case of AS. Therefore, while NbutGT could be beneficial in treating septic shock, it may not be as effective in AS curative treatment, given that the NbutGT-induced twofold increase in O-GlcNAcylation requires at least one hour.

Concerning hemodynamics, we showed that the adjuvant pretreatment with NbutGT did not modify the blood pressure response to epinephrine with no difference in HR between the AS+EPI+preNBut and AS+EPI groups throughout the experiment. As our epinephrine administration protocol was based on a target blood pressure value, we also did not observe any difference in terms of cumulative epinephrine dose.

On the other hand, we observed a significant enhancement in cardiac function (contractility and cardiac output) with the association between NbutGT and epinephrine. Pretreatment with NbutGT did not modify the initial response to epinephrine with regard to LVSF and CO; however, from T35 and T40 min, we observed an improvement in LVSF and CO, respectively, with the association between epinephrine and NbutGT (AS+EPI+preNBut group) as compared to epinephrine alone (AS+EPI group). Tajima et al. showed that the association of epinephrine and volume expansion with hydroxyethyl starch fluid was the most effective in improving MAP and CO at the initial resuscitation of AS [22]. Ferron et al. observed in a model of septic shock that stimulation of O-GlcNAcylation by NbutGT was associated with improved function in several organs (heart, kidneys), a decrease in blood lactate level and a reduction in mortality. Interestingly, these authors observed that NbutGT increased LVSF but had no significant effect on MAP, depicting hemodynamic and cardiovascular modulation similar to that which we observed [29]. In our work, NbutGT administration led to a later response to epinephrine with regard to LV contractility and CO. It suggests that NbutGT may preferentially enhance the metabolism and contractility of cardiomyocytes by increasing O-GlcNAcylation, thereby improving cardiac performance. However, the impact of NbutGT on vascular smooth muscle cells may be less suitable for increasing mean arterial pressure (MAP). Therefore, potential clinical implications could involve improving myocardial oxygen utilization during AS, reducing cardiac dysfunctions and preventing epinephrine-refractory cases of AS.

To further investigate the underlying mechanisms and describe cardiac function, we noted an increase in CO primarily associated with the increase in LVEDD, as LVESD was not significantly different between the epinephrine-treated groups. Consequently, this suggests that the elevated CO results mainly from enhanced SV and probably cardiac filling, potentially due to an augmented venous return and better cardiac relaxation. Indeed, although our study did not provide sufficient evidence to identify such specific parameters, O-GlcNAcylation enhanced vascular contractile response [30] and Silva et al. reported during septic shock that an acute increase in O-GlcNAcylation by OGT inhibition reduced systemic inflammation and attenuated hypotension and the vascular refractoriness to phenylephrine, improving survival [31]. Additionally, the vascular hyperpermeability of anaphylactic shock could be reduced, partially explaining the increase in preload and LVEDD.

Finally, in this study, we found that NbutGT, unlike epinephrine alone, significantly improved cardiac mitochondrial respiration. Indeed, mitochondrial chain complex activities were restored and OXPHOS was enhanced through complex II at T15 min and through complexes I and II at T60 min. Pretreatment with NbutGT, either alone or in combination with epinephrine, demonstrated a protective effect on mitochondrial respiration. Our study showed that mitochondrial respiratory chain improvement was likely associated with cellular protective mechanisms against acute injury. Nowadays, only a few studies have elucidated the mechanisms by which O-GlcNAcylation levels modulate mitochondrial function during situations of acute stress. After an increase in cardiac O-GlcNAcylation level, Champattanachai et al. observed that mitochondrial redistribution of the anti-apoptotic Bcl-2 proteins protected cardiomyocytes from acute ischemia–reperfusion injury [32]. The underlying mechanisms by which NbutGT restores mitochondrial respiratory activity are not fully understood, since the repertoire of proteins affected as well as the functional impact remains ill-defined. However, O-GlcNAcylation can affect mitochondrial function in different pathways:-A recent study [15] that compared in vitro reconstitution assay and in vivo experiments showed that O-GlcNAcylation cycling enzymes are present in functionally relevant amounts in the mitochondrial compartment and can trigger rapid changes in protein O-GlcNAcylation. The authors proposed that mitochondrial protein’s hyper-O-GlcNAcylation led to the enhancement of their enzymatic activity to increase mitochondrial respiratory and decrease ROS release.-NbutGT can decrease SERCA2a protein overexpression at the early phase of septic shock to maintain calcium homeostasis in mitochondria [29].-NbutGT can also attenuate mPTP opening via O-GlcNAcylation of several subunits and, thus, can prevent loss of mitochondrial membrane potential or avoid mitochondrial swelling in the context of ischemia–reperfusion injury [33].

Mitochondrial improvement might also participate in better cardiac function, including a better relaxation that needs mitochondrial-driven energy likely resulting in enhanced cardiac filling.

At the same time, we observed a more marked decrease in blood lactate in the AS+EPI+preNBut group as compared to the AS+EPI group. NbutGT administration may influence overall metabolic homeostasis by regulating different enzymatic pathways, potentially explaining the decrease in lactatemia. Lactate production might be reduced by NbutGT through the activation of glycolysis-related key enzymes via O-GlcNAcylation. Additionally, the decrease in lactatemia could be associated with improved mitochondrial respiration as observed in our work, resulting from an increased demand for substrates such as pyruvate or lactate itself [34]. Finally, O-GlcNAcylation could enhance the Cori cycle, facilitating the recycling of lactate by hepatocytes. To our knowledge, no study has evaluated the effects of NbutGT and O-GlcNAcylation on these three metabolic pathways. The relationship between lactate, NbutGT and O-GlcNAcylation is likely much more intricate than we were able to explore in this study, particularly considering the complex metabolism of lactate and the crucial role of O-GlcNAcylation in regulating numerous biological processes. Ensuring that blood lactate levels decrease is a therapeutic objective in itself, notably in the context of septic shock and severe ischemia–reperfusion settings. This is also an important beneficial effect of NbutGT [18,35].

In brief, we observed with the epinephrine–NbutGT association an increase in LV systolic function assimilated to an increase in cardiac contractility and CO, which we have already described elsewhere [4,22,36]. This improvement in cardiac performance was associated with a simultaneous increase in cardiac mitochondrial respiration and a decrease in blood lactatemia. A straightforward line of thought would be to consider that improving the respiratory chain could both increase ATP production and therefore maintain a high level of cardiac contraction, and also improve glycolysis/mitochondrial coupling and therefore, ultimately, decrease blood lactate levels.

To explore the potential inclusion of NbutGT in AS therapy, further studies are required to characterize the long-term impact of NbutGT administration in pretreatment therapy and to assess the optimal dose, time point and duration of treatment in order to construct the most effective therapeutic regimen. Actually, the number of in vivo studies is quite limited and only focuses on murine models. A clinical study showed that the O-GlcNAcylation level increase was responsible for a cardioprotective effect through a preconditioning mechanism [37]. Hence, it remains an important step to continue toward clinical validation. The beneficial effect of NbutGT administration is already established in several acute injuries (trauma–hemorrhage shock, septic shock or ischemia–reperfusion). The studies described underlying mechanisms such as improved mitochondrial function and a reduction in cell apoptosis.

Concerning the translational potential of the findings, there is insufficient evidence to justify testing NbutGT in clinical trials. The only effective treatment remains epinephrine, known for its immediate and potent systemic effects. Our team has already tested various treatments and signaling pathways to enhance AS treatment. Post-treatment with methylene blue (MB) or ABT-491 (a platelet-activating factor inhibitor) improves the survival of animals and reduces the dose of epinephrine infused. Further clinical studies could explore a combination of NbutGT pretreatment with different post-treatment adjuvants for AS management. The potential benefit of NbutGT pretreatment could be to prevent a severe anaphylactic reaction in a patient with a known allergy, especially when the reintroduction of the allergen is inevitable. Actually, there are no reported side effects or contraindications associated with the use of NbutGT [18], but further studies are required to confirm the safety of NbutGT administration.

Our study has several limitations. First, we did not investigate cardiac output or cardiac contractility with invasive methods, which would have supported a more accurate evaluation of the hemodynamic and cardiovascular profiles of this study. Nevertheless, we had measured cardiac output in several other studies using the same experimental model and our model has always shown high reproducibility of cardiac hemodynamic parameters. Moreover, invasive methods, such as the measurement of maximum d*P*/d*t*, require the insertion of a probe into the left heart ventricle that can induce myocardial injuries, thus biasing the evaluation of cellular lesions directly related to the AS itself. Second, an O-GlcNAcylation level variation through aging should be considered prior to determining the beneficial effect of NbutGT. Cardiac metabolism is subject to modifications throughout aging [38]: at the early stage of life, substrates for energy production are carbohydrates (glucose, lactate and pyruvate), and, in the adult heart, energy is mainly supplied by fatty acid oxidation. Therefore, basal O-GlcNAcylation levels may vary depending on the age with a decrease in O-GlcNAcylation levels in rat hearts between 6 and 22 weeks [39] or an increase in O-GlcNAcylation levels in rat hearts between 5 and 24 months [40]. Further studies are needed to explore the link between O-GlcNAcylation levels age-associated variations and NbutGT effects in the field of anaphylaxis.

Overall, we have identified a metabolic pathway associated with the use of NbutGT that may be relevant in the context of AS. It could be beneficial to prevent a severe anaphylactic reaction in a patient known to be allergic and in whom allergen re-exposure is inevitable.

To our knowledge, our work is the first to suggest that O-GlcNAcylation may have a protective effect in a model of AS treated with epinephrine.

## 4. Materials and Methods

### 4.1. Animals

All animal procedures and care were performed in accordance with the European Communities Council Directive of 24 November 1986 (86/609/EEC). The study protocol was approved by the Ethical Committee for Animal Experiments of Strasbourg University, France (N° #16987-2018100500047648).

Ten-week-old male Brown Norway rats (Charles River laboratories, Saint-Germain-Nuelles, France) weighing 250–300 g were sensitized by subcutaneous administration of chicken egg albumin (ovalbumin, OVA), as previously described [21,41]. On days 0, 7 and 14, sensitization was performed by subcutaneous injection of 1 mg OVA (Sigma-Aldrich, Merck laboratory, Taufkirchen, Germany) and 4 mg aluminum hydroxide (Al(OH)_3_, Sigma-Aldrich, Merck laboratory, Germany) as adjuvant diluted in 1 mL 0.9% saline solution. Animals were challenged on day 21. AS was induced by an intravenous (IV) injection of 1 mg OVA diluted in 1 mL of 0.9% saline solution (T0) in all shocked groups. In the control group, 1 mL of 0.9% saline solution was injected (T0). This model of AS was largely described by different groups including ours. Brown Norway rats were considered the most suitable strain for inducing specific IgE responses [42,43]. Age was considered to minimize variability in IgE production prior to the weaning period. The inclusion of the male gender was based on the known disparity in immune responses between males and females [44]. However, further studies will be needed to determine the female response in such a setting.

### 4.2. Surgical Preparation

Anesthesia was induced with 5% isoflurane (Forène, Abbott, Rungis, France) and maintained with 1% isoflurane, as previously described. Isoflurane may impact hemodynamics measurements probably by inhibiting cardiac cell death and improving cardiac mitochondrial respiration. However, it was used in both groups to avoid confusion biases. The trachea was intubated, and the lungs were mechanically ventilated (Alpha Lab, Equipement Vétérinaire Minerve, Esternay, France) with 100% FiO_2_. Respiratory parameters were a tidal volume of 2.2 mL and a respiratory rate of 60/min during the stabilization period. A fluid-filled polyethylene catheter was inserted into the left femoral vein for fluid maintenance (10 mL/kg/h of 0.9% saline solution) or fluid resuscitation (30 mL/kg/h of 0.9% saline solution) and administration of OVA in the shocked groups. At the same time, a fluid-filled polyethylene catheter was inserted into the right femoral artery to measure hemodynamic parameters. At the end of the experiment and after blood sampling, rats were euthanized with isoflurane and the left ventricles were excised.

### 4.3. Study Design

We conducted a two-step study. Firstly, we compared two time points of NbutGT administration (pretreatment and post-shock) by assessing the level of O-GlcNAcylation of left ventricular proteins. Secondly, we studied the hemodynamic, cardiac and mitochondrial effects of the therapeutic combination of epinephrine and NbutGT as pretreatment therapy, based on first-step results, during anaphylactic shock.

### 4.4. First-Step Study: Effect of NbutGT on O-GlcNAcylation of Left Ventricle Proteins as a Function of Administration Regimens

In this first part of the study, samples of the left ventricle were taken at T15 min in all groups. We studied two NbutGT administration regimens, comparing the level of O-GlcNAcylation, OGT/OGA expression and cardiac mitochondrial respiration in the control, AS, AS+pre-Nbut and AS+post-Nbut groups. After a 15 min period of stabilization, sensitized rats were randomly allocated to the following groups (n = 10):-Control group at T15 (CON-T15): We used a specific control group for which 1 mL of 0.9% saline solution was administrated at T0, followed by continuous infusion (10 mL/kg/h) and one bolus of 1 mL 0.9% saline (T3).-Anaphylactic shock group (AS): Injection of OVA diluted in 1 mL of 0.9% saline (T0), followed by continuous infusion of 0.9% saline solution (30 mL/kg/h) and one bolus of 1 mL 0.9% saline (T3).-NbutGT as a pretreatment (AS+pre-Nbut): NbutGT (10 mg/kg diluted in 1 mL 0.9% saline solution) was administrated by IV injection in the penile vein 1h before the injection of OVA (T0). Fluids management was the same as in the AS group.-NbutGT as a post-AS therapy (AS+post-Nbut): NbutGT (10 mg/kg diluted in 1 mL 0.9% saline solution) was administrated by IV injection in the penile vein 3 min (T3) after the injection of OVA (T0). Then, fluids management was the same as in the AS group.

### 4.5. O-GlcNAcylation Level and OGT/OGA Expression

Western blotting experiments were performed on left ventricle (LV) tissue samples, as previously described [29], at T15 min in the CON-T15, AS, AS+pre-Nbut and AS+post-Nbut groups. OGT and OGA are described as cycling enzymes for their role in regulating O-GlcNAcylation: OGT adds the modification and OGA removes it. Proteins were evaluated with a Western blot analysis using the following antibodies: O-GlcNAcylation (abcam ab201995, dilution 1/20,000, Cambridge, UK), OGT (cell signaling 24083, dilution 1/400, Danvers, MA, USA) and OGA (Abcam 105217, dilution 1/5000). Analysis was performed using Image Lab software (Image Lab 6.1, Bio-Rad, Hercules, CA, USA).

### 4.6. Second-Step Study: Effects of NbutGT Pretreatment on Hemodynamics, Cardiac Function and Mitochondrial Respiration

In this second part of the study, samples of the left ventricle were taken at T15 min in the AS group and at T60 min in all other groups. After a 15 min period of stabilization, sensitized rats were randomly allocated to the following groups (n = 10):-Control group (CON): 1 mL of 0.9% saline solution administrated at T0 followed by continuous infusion (10 mL/kg/h) with two boluses of 1 mL 0.9% saline (T3 and T5).-Anaphylactic shock (AS): Injection of OVA diluted in 1 mL of 0.9% saline (T0), followed by continuous infusion of 0.9% saline solution (30 mL/kg/h) and two boluses of 1 mL 0.9% saline (T3 and T5).-AS+EPI: Same procedure as for AS group + epinephrine treatment: two boluses of epinephrine (Adrenaline, Aguettant) 2.5 μg diluted in 1 mL 0.9% saline at T3 and T5. Then, a continuous infusion of epinephrine (10 μg/kg/min) was started at T5 as described previously [4]. Epinephrine continuous infusion was adjusted by 0.5 μg/kg/min every five minutes to target a mean arterial blood pressure (MAP) of 100 mmHg.-AS+EPI+preNBut: Same procedure as for the AS+EPI group + pretreated with NbutGT; NbutGT (10 mg/kg diluted in 1 mL 0.9% saline solution) was administrated by IV injection in the penile vein 1h before the injection of OVA (T0).

The evaluation times for the hemodynamic and cardiac function parameters were T0 (just before OVA injection), T1, T2.5, T5 and then every 2.5 min until the end of the experiment (T15 for the AS group and T60 min for the other groups). The dosage of NbutGT at 10 mg/kg was determined based on the studies conducted by Macauley et al. [19,28], where the authors investigated different concentrations of NbutGT to inhibit O-GlucNAcase (OGA). Their findings established a dose–response relationship between NbutGT concentration and the increase in O-GlcNAcylation levels. Furthermore, Ferron et al. [29] recommended the use of NbutGT at 10 mg/kg as the optimal dose to increase cardiac protein O-GlcNAcylation in an experimental model of septic shock.

In our study, the timing of NbutGT administration was determined to reflect clinically relevant situations, especially during the perioperative period known at higher risk for AS occurrence. Furthermore, it is well established that O-GlcNAcylation levels exhibit a time-dependent increase within the first hour after administration, achieving two-fold elevation over the basal level [19].

### 4.7. Hemodynamic Parameters

Mean arterial blood pressure (MAP) and heart rate (HR) were monitored through the arterial catheter inserted into the right femoral artery. MAP and HR were measured continuously with LabChart 7 software (ADInstrument, Sydney, Australia).

The total dose of epinephrine required to comply with the epinephrine administration protocol was measured in the two groups treated with epinephrine (AS+EPI and AS+EPI+preNBut).

### 4.8. Cardiac Function

Cardiac function was studied by assessing contractility and cardiac output (CO).

Contractility of the left ventricle was evaluated by the left ventricular shortening fraction (LVSF) calculated by the LVEDD and the LVESD, as follows: LVSF = (LVEDD − LVESD)/LVEDD × 100. Prior to performing an echocardiography, the thorax of rats was depilated. After surgical preparation, LVEDD and LVESD were assessed using an ultrasound system (Vivid 7, GE Healthcare, Velizy, France) equipped with a 12 MHz sectorial probe. The image was considered valid if the left ventricle appeared circular with a visualization of the two papillary muscles in the short-axis view.

Cardiac output (CO) can be calculated using the formula, CO = SV × HR, where SV represents stroke volume and HR represents heart rate. Stroke volume (SV) is determined by the left ventricle end-diastolic volume (LVEDV) and left ventricle end-systolic volume (LVESV), which are in turn estimated using the Teichholz method based on left ventricle end-diastolic diameter (LVEDD) and left ventricle end-systolic diameter (LVESD), as previously described [20].

### 4.9. Mitochondrial Respiration

Mitochondrial function of the heart was assessed by oximetry. At the end of the experiments (T15 min in the first-step study and T60 min in the second-step study), the left ventricle was extracted. The mitochondrial respiratory chain was assessed with a high-resolution respirometer (Oxygraph-2 k; Oroboros Instruments, Innsbruck, Austria), with continuous stirring at 37 °C in a buffer. Briefly, as previously described [11], fibers (1 mg wet weight) were separated and then permeabilized in a bath of solution containing 50 μg/mL saponin for 30 min at 4 °C with shaking. A multiple substrate-inhibitor titration protocol was used for the analysis of non-phosphorylating respiration and oxidative phosphorylation. In total, 5 mM glutamate and 2 mM malate were used to determine non-phosphorylating respiration initially supported by complex I (CI-linked substrate state). The second step was the injection of ADP (2 mM) at a saturating concentration, which activates the ATP synthase, and oxidative phosphorylation (OXPHOS) was stimulated through complex I to obtain CI OXPHOS state. Then, 25 mM succinate, which activates complex II, was added to obtain the maximum OXPHOS coupling state or CI + II OXPHOS state. Finally, complex I was blocked with rotenone (0.02 mM) to give the OXPHOS state initiated only through complex II (CII-linked OXPHOS state). The oxygen consumption was analyzed using DatLab software 4.3. Data are expressed as O_2_ pmol/s/mg wet weight.

### 4.10. Lactatemia

Using a lactate Scout+ (EKF diagnostics, Cardiff, UK), blood lactate level was measured at T0 and at the end of the experiment through the arterial catheter inserted into the right femoral artery.

### 4.11. Statistical Analysis

Values were expressed as mean ± standard error of the mean (SEM). Gaussian distributions were assessed by using Shapiro–Wilk’s test. If the distribution of the samples respected a Gaussian law, parameters were compared between groups using ANOVA; otherwise, the comparison was performed using a Kruskal–Wallis test. Post hoc analyses were performed for the significant variables in ANOVA to determine which groups differed from each other, using Tukey’s multiple comparisons test. A linear mixed model was used to evaluate the evolution of variables over time for each group. Because of the potential for type I error due to multiple comparisons, results should be interpreted as containing some biases. However, as previously observed in experimental studies, 5 to 10 rats per group are required to demonstrate, at least, a 30% reduction in cardiac and mitochondrial functions in our model of AS [4,11,22]. A *p*-value < 0.05 was considered statistically significant. Statistical analyses were performed using Prism 8.4.0 (Graph Pad Software Inc., San Diego, CA, USA).

## 5. Conclusions

Pretreatment with NbutGT before anaphylactic shock increases the O-GlcNAcylation of cardiac proteins without altering the OGA and OGT expression, suggesting a post-transcriptional process. Pretreatment with NbutGT modulates the cardiac response to epinephrine in a diverse pattern: an increase in contractility and cardiac output. These modified cardiac functions may be linked to the partial restoration of the respiratory chain complexes activity in cardiac mitochondria. Finally, pretreatment with NbutGT is associated with a significant reduction in blood lactate levels. The elevation of O-GlcNAcylation in cardiac proteins seems to confer protective effects in our model of AS.

## Figures and Tables

**Figure 1 ijms-25-03316-f001:**
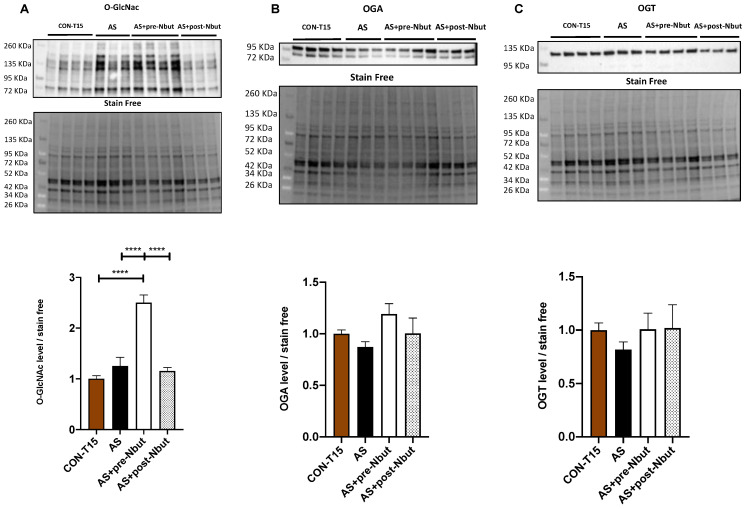
Western blot analysis. (**A**) O-GlcNAcylation levels, (**B**) OGA, and (**C**) OGT expression in the heart. CON-T15: control group at T15 min (n = 10); AS: anaphylactic shock group (n = 7); AS+pre-Nbut: AS pretreated with NButGT group (n = 10); AS+post-Nbut: NButGT treatment after AS (n = 10). Quantification was performed in relation to stain free method. Values are means ± SEM, **** *p* < 0.0001.

**Figure 2 ijms-25-03316-f002:**
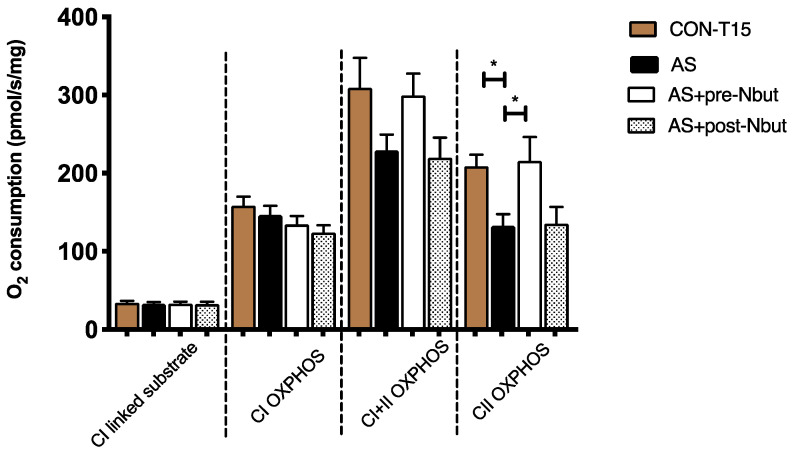
Respiratory chain complexes enzymatic activity between groups in the first-step study. CI: mitochondrial complex I; CI + II: mitochondrial complexes I and II; OXPHOS: mitochondria ADP-activated state of oxidative phosphorylation; CII: mitochondrial complex II. CON-T15: control group at T15 min (n = 10); AS: anaphylactic shock group (n = 7); AS+pre-Nbut: AS pretreated with NButGT group (n = 10); AS+post-Nbut: NButGT treatment after AS (n = 10). Values are means ± SEM, * *p* < 0.05.

**Figure 3 ijms-25-03316-f003:**
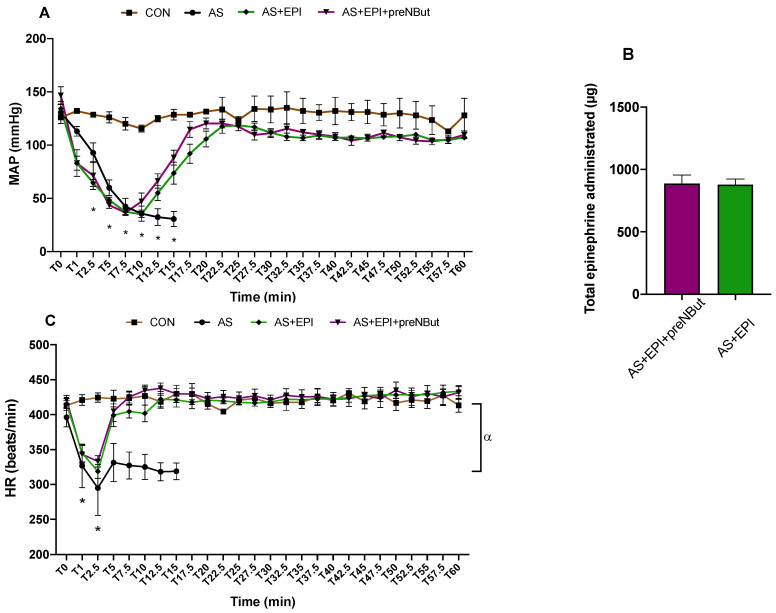
Hemodynamic parameters. (**A**) Time course of systemic mean arterial pressure (MAP). (**B**) Total epinephrine dose infused during the experiment. (**C**) Time course of heart rate (HR). CON: control group; AS: anaphylactic shock group; AS+EPI: AS treated with epinephrine group; AS+EPI+preNBut: AS pretreated with NButGT and treated with epinephrine group. N = 10 in each group. Data are presented in means ± SEM. * MAP and HR decreased significantly in AS, AS+EPI and AS+EPI+preNBut groups as compared with CON group (*p* < 0.01). *α*: HR decreased significantly in AS group as compared with CON group from T0 to T15 min (*p* < 0.0001).

**Figure 4 ijms-25-03316-f004:**
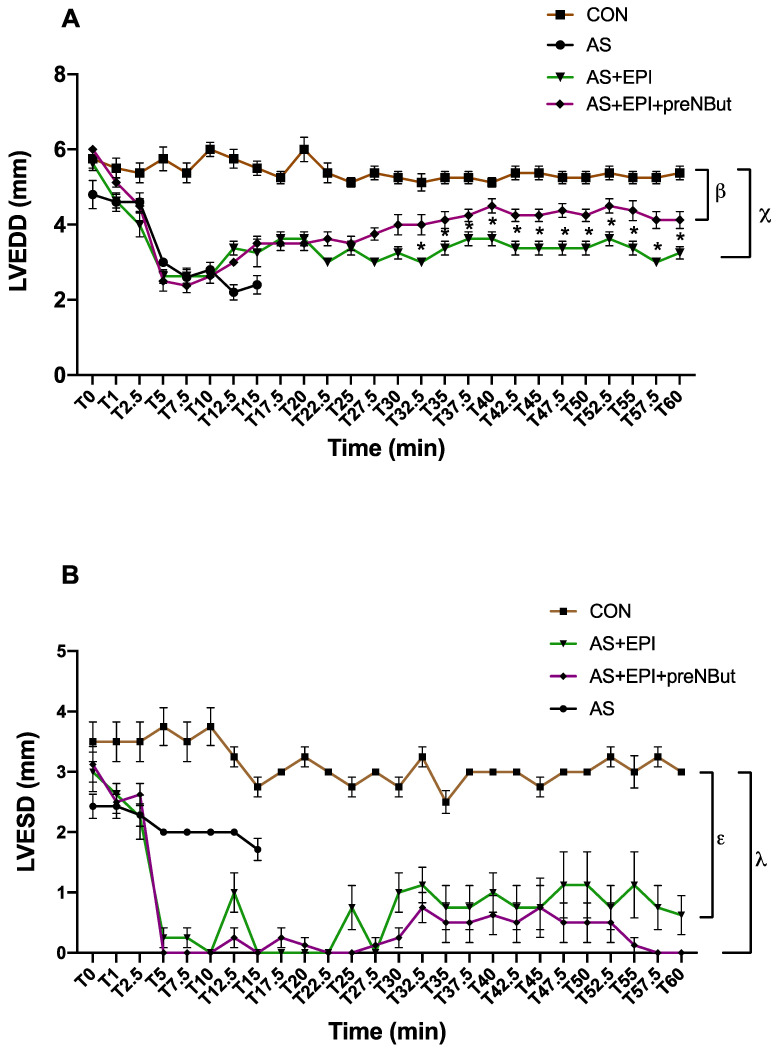
Cardiac function. (**A**) Time course of left ventricle end-diastolic diameter (LVEDD). β LVEDD time course profile significantly lower in the AS+EPI+preNBut group compared to the CON group from T5 to T60 min, *p* < 0.05. χ LVEDD time course profile significantly lower in the AS+EPI group compared to the CON group from T5 to T60 min, *p* < 0.0001. *: *p* < 0.05 between-group differences AS+EPI+preNBut group vs. CON group. (**B**) Time course of left ventricle end-systolic diameter (LVESD). ε LVESD time course profile significantly lower in the AS+EPI group compared to the CON group from T5 to T60 min, *p* < 0.0001. λ LVESD time course profile significantly lower in the AS+EPI+preNBut group compared to the CON group from T5 to T60 min, *p* < 0.0001. CON: control group; AS: anaphylactic shock group; AS+EPI: AS treated with epinephrine group; AS+EPI+preNBut: AS pretreated with NButGT and treated with epinephrine group. N = 10 in each group. T0: ovalbumin injection.

**Figure 5 ijms-25-03316-f005:**
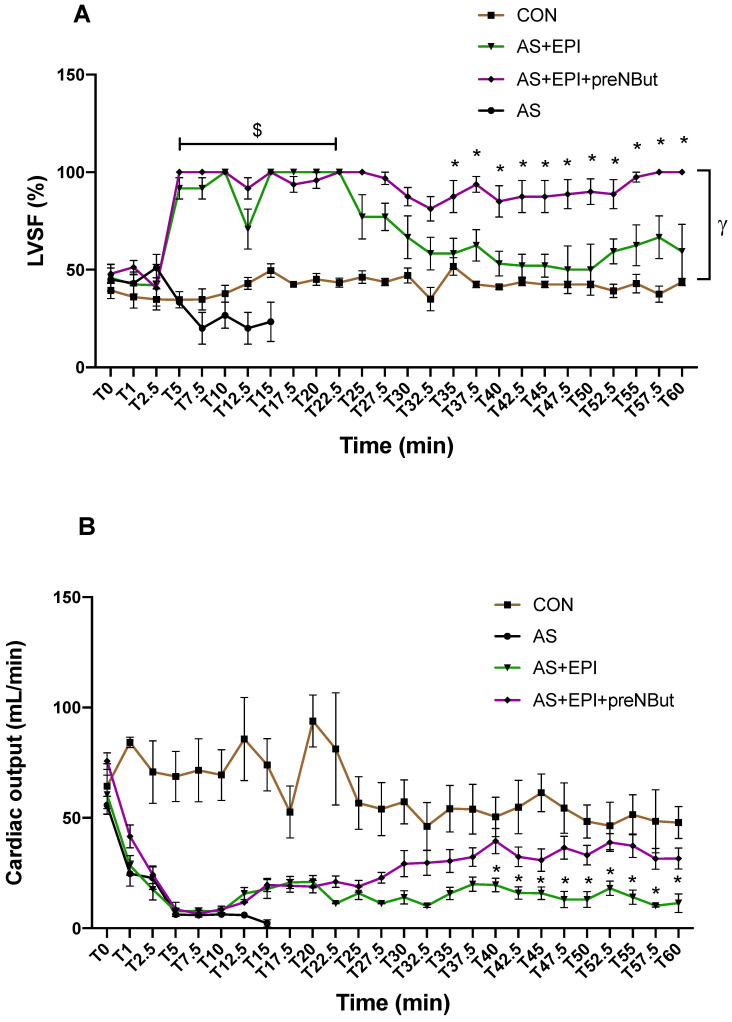
Cardiac function. (**A**) Time course of left ventricular shortening fraction (LVSF). $ *p* < 0.05 between-group differences AS+EPI group vs. CON group. * *p* < 0.05 between-group differences AS+EPI+preNBut group vs. AS+EPI group. γ: LVSF time course profile significantly higher in the AS+EPI+preNBut group compared to the CON group from T5 to T60 min, *p* < 0.01. (**B**) Time course of cardiac output (CO). * *p* < 0.05 between-group differences AS+EPI+preNBut group vs. AS+EPI group. CON: control group; AS: anaphylactic shock group; AS+EPI: AS treated with epinephrine group; AS+EPI+preNBut: AS pretreated with NButGT and treated with epinephrine group. N = 10 in each group. T0: ovalbumin injection.

**Figure 6 ijms-25-03316-f006:**
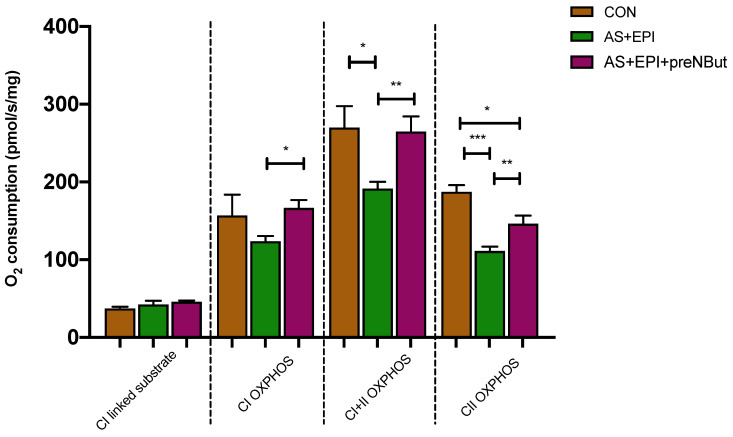
Respiratory chain complexes enzymatic activity between groups. CI: mitochondrial complex I; CI + II: mitochondrial complexes I and II; OXPHOS: mitochondria ADP-activated state of oxidative phosphorylation, CII: mitochondrial complex II. CON: control group; AS+EPI: AS treated with epinephrine group; AS+EPI+preNBut: AS pretreated with NButGT and treated with epinephrine group. N = 10 in each group. Values are means ± SEM, * *p* < 0.05 ** *p* < 0.01 *** *p* < 0.01.

**Figure 7 ijms-25-03316-f007:**
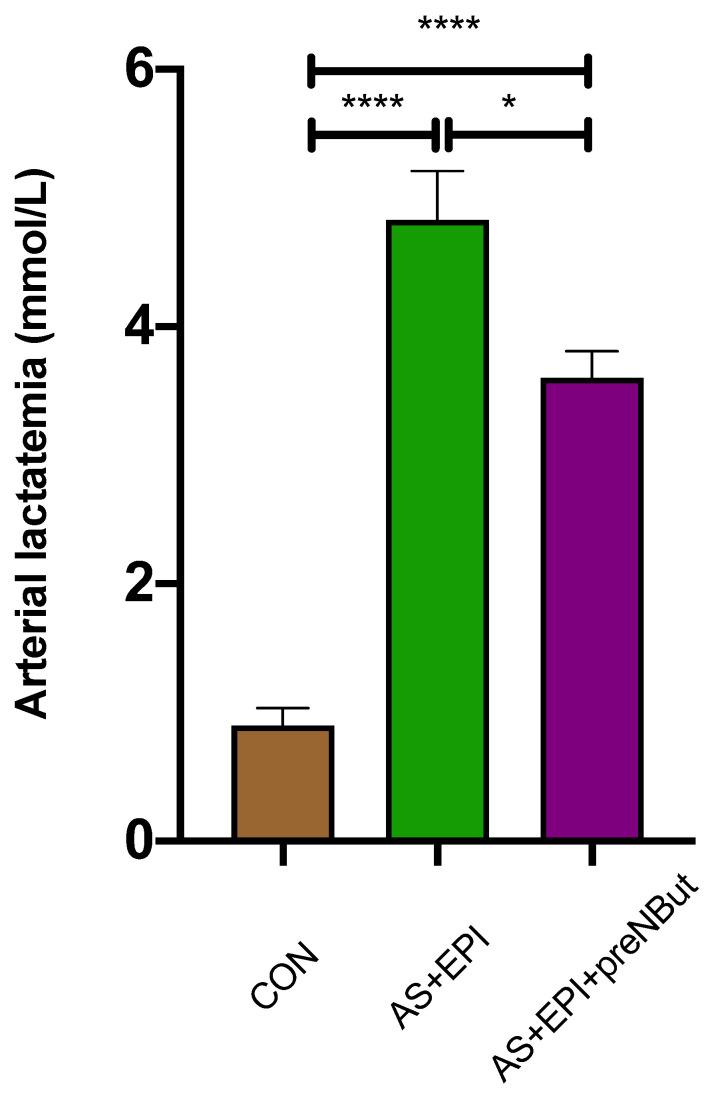
Lactatemia at the end of experiment (T60 min in CON, AS+EPI and AS+EPI+NB groups). CON: control group; AS+EPI: shocked group treated with epinephrine; AS+EPI+preNBut: shocked group treated with NButGT before shock induction and epinephrine. N = 10 in each group. Data are presented in means ± SEM, * *p* < 0.05 **** *p* < 0.0001.

**Table 1 ijms-25-03316-t001:** Mitochondrial complexes activity rates (O_2_ pmol/s/mg) between groups. CON: control group; AS+EPI: AS treated with epinephrine group; AS+EPI+preNBut: AS pretreated with NButGT and treated with epinephrine group. N = 10 in each group. Values are means ± SEM. ^&^ *p* < 0.05 vs. AS+EPI; ^&&^ *p* < 0.01 vs. AS+EPI; ^#^ *p* < 0.5 vs. AS+EPI; ^##^ *p* < 0.001 vs. AS+EPI; ^@^ *p* < 0.05 vs. AS+EPI+preNBut.

	CON	AS+EPI	AS+EPI+preNBut
CI linked substrate	37 (2)	42 (5)	46 (2)
CI OXPHOS	157 (27)	124 (7)	171 (11) ^&^
CI+II OXPHOS	270 (26) ^#^	191 (9)	270 (21) ^&&^
CII OXPHOS	187 (9) ^##, @^	111 (5)	150 (11) ^&&^

## Data Availability

Data are contained within the article.

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
