# Peer review of "NButGT Reinforces the Beneficial Effects of Epinephrine on Cardiac Mitochondrial Respiration, Lactatemia and Cardiac Output in Experimental Anaphylactic Shock"

_ijms, 2024, doi:10.3390/ijms25063316_

Round 1

Reviewer 1 Report

Comments and Suggestions for Authors

Your manuscript offers insightful research into the use of NButGT in anaphylactic shock management, particularly concerning improved cardiac outcomes. To further strengthen your work, please consider providing a more detailed rationale for the choice of your animal model and the NButGT dosing. The potential cardiovascular influence of isoflurane anesthesia on your results would also merit discussion. Additional clarity on the post-transcriptional mechanisms suggested by your findings and the clinical translational potential of NButGT pretreatment could enhance the manuscript. Addressing these points will solidify the relevance and impact of your research within the field.

Comments:

Study Design and Animal Model:

  1. Animal Model Relevance: The use of Brown Norway rats sensitized with ovalbumin is a well-established model for studying anaphylactic shock. It would be beneficial if the authors provided justification for the choice of this particular strain and age of rats, as well as any gender effects, since male rats were exclusively used.
  2. Ethical Compliance: The study adheres to the European Communities Council Directive, which is commendable. The approval by the Ethical Committee for Animal Experiments adds to the study's ethical robustness.

Experimental Protocols:

  1. Anesthesia and Surgical Technique: The use of isoflurane and mechanical ventilation is appropriate for such experiments. However, the potential cardiovascular effects of isoflurane should be acknowledged as it may impact the study's hemodynamic measurements.
  2. Dosing and Timing: The dosing of NButGT and the timing of its administration are critical factors. A rationale for the chosen dose and its timing relative to the induction of AS would strengthen the study. Moreover, it would be instructive to know whether there is a dose-response relationship.

Results and Data Interpretation:

  1. O-GlcNAcylation: The authors demonstrate that NButGT increases O-GlcNAcylation levels without altering OGA and OGT expression, which they claim suggests a post-transcriptional mechanism. Clarification on how this conclusion was reached and whether alternative mechanisms were considered would be valuable.
  2. Hemodynamic Parameters: The results indicate that NButGT does not alter the initial hemodynamic response to epinephrine, but it does improve cardiac function over time. It would be useful if the authors discussed the potential clinical implications of these findings.
  3. Mitochondrial Function: The study provides evidence that NButGT restores mitochondrial respiratory chain complex activity. The mechanisms by which NButGT exerts these effects warrant a more in-depth discussion. Are there specific mitochondrial adaptations or changes in substrate utilization that could explain the observed improvements?
  4. Lactatemia: The significant reduction in lactate levels with NButGT pretreatment is a notable finding. However, the discussion could benefit from a deeper exploration of how this might affect overall metabolic homeostasis beyond the scope of lactate levels.

Methodological Considerations:

  1. Statistical Analysis: While the study uses appropriate statistical methods, the article should discuss any potential limitations or biases in the statistical analysis, such as sample size considerations, the use of multiple comparisons, and the potential for type I and II errors.
  2. Reproducibility: To ensure reproducibility, the authors should provide sufficient details about the study's protocols. Any proprietary or unique methodologies should be described in detail or made available upon reasonable request.

Broader Context and Implications:

  1. Clinical Relevance: The translational potential of the findings should be addressed. How might NButGT pretreatment be integrated into current clinical protocols for managing AS? Are there known side effects or contraindications?
  2. Future Directions: The study opens avenues for future research. The authors should propose next steps, which could include investigation into the long-term effects of NButGT pretreatment, potential human clinical trials, and exploration of NButGT’s impact on other forms of shock.

Overall Assessment:

The study presents novel findings that could have significant implications for the treatment of anaphylactic shock. The authors have made a compelling case for the benefits of NButGT pretreatment. However, for a more comprehensive understanding of NButGT's role, additional studies are required to elucidate the exact mechanisms, potential long-term effects, and how these findings translate to human models. The authors are encouraged to address the points above to strengthen their arguments and provide a clearer pathway for clinical application.

Reviewer 2 Report

Comments and Suggestions for Authors

This manuscript reveals the beneficial effects of NButGT as a promoter of O-GlcNAcylation in anaphylactic shock. It demonstrated that pretreatment with NButGT increased O-GlcNAcylation of cardiac proteins and had an additive effect with epinephrine, improving cardiac output and mitochondrial respiration while decreasing blood lactate levels. The data in the article are robust and clinically relevant. However, I still have the following questions:

  1. 1. My primary concern is whether there are differences between the NButGT pre-treatment group and post-treatment group in terms of cardiac and mitochondrial function in the first step experiment. Additionally, in Ferron et al. study (doi.org/10.1038/s41598-019-55381-7), NButGT was administered after 3 hours of septic shock, yet it still increased O-GlcNAcylation levels. What are the differences between these two types of shock? Please provide explanations in the discussion.

  2. 2. The number of rats involved in the animal experiments is not clearly stated. The Methods section only mentions n=10, but it is unclear whether this refers to each group or the total number. Please add annotations in the figure legends indicating the number of rats in each group.

  3. 3. Please analyze the limitations of the experiment in the Discussion section.

Round 2

Reviewer 1 Report

Comments and Suggestions for Authors

The authors have answered my queries satisfactorily.

Reviewer 2 Report

Comments and Suggestions for Authors

Thank authors for response and revisions. The manuscript has made significant progress. The addition of mitochondrial function data of both pre-treatment and post-treatment groups enriched the workload and showed that NBUT should be administered before AS. Furthermore, the Discussion Part now includes explanations of different shock models, mitochondrial function results, and lactate levels, resulting in improved logic and emphasizing the novelty of the research. I believe this manuscript is suitable for publication in IJMS.